# Efficient De Novo Biosynthesis of Heme by Membrane Engineering in *Escherichia coli*

**DOI:** 10.3390/ijms232415524

**Published:** 2022-12-08

**Authors:** Zhexian Geng, Jinxia Ge, Wei Cui, Hui Zhou, Jieying Deng, Baocai Xu

**Affiliations:** 1School of Food and Biological Engineering, Hefei University of Technology, Hefei 230601, China; 2Engineering Research Center of Bio-Process, Ministry of Education, Hefei University of Technology, Hefei 230601, China

**Keywords:** 5-aminolevulinic acid, heme, *Escherichia coli*, membrane engineering, metabolic engineering

## Abstract

Heme is of great significance in food nutrition and food coloring, and the successful launch of artificial meat has greatly improved the application of heme in meat products. The precursor of heme, 5-aminolevulinic acid (ALA), has a wide range of applications in the agricultural and medical fields, including in the treatment of corona virus disease 2019 (COVID-19). In this study, *E. coli* recombinants capable of heme production were developed by metabolic engineering and membrane engineering. Firstly, by optimizing the key genes of the heme synthesis pathway and the screening of hosts and plasmids, the recombinant strain EJM-pCD-AL produced 4.34 ± 0.02 mg/L heme. Then, the transport genes of heme precursors *CysG*, *hemX* and *CyoE* were knocked out, and the extracellular transport pathways of heme Dpp and Ccm were strengthened, obtaining the strain EJM-ΔCyoE-pCD-AL that produced 9.43 ± 0.03 mg/L heme. Finally, fed-batch fermentation was performed in a 3-L fermenter and reached 28.20 ± 0.77 mg/L heme and 303 ± 1.21 mg/L ALA. This study indicates that *E. coli* recombinant strains show a promising future in the field of heme and ALA production.

## 1. Introduction

Heme is a stable form of iron containing a porphyrin compound (C_34_H_33_FeN_4_O_4_) that can interact with biological membranes, existing in almost all kinds of animals, and participating in a variety of physiological and biochemical reactions including respiration, cell differentiation and signal transduction [1,2,3,4]. Heme has a strong coloring ability that can replace coloring agents and synthetic pigments in food, reduce or completely replace the use of nitrite, reduce the residue of nitrite in food and prevent food from forming nitrosamines and other carcinogens [5,6]. In the past several years, heme has been used as a good iron supplement and has shown a significant therapeutic effect in the treatment of iron-deficiency anemia [7,8]. At the same time, the heme iron complex is one of the most important cofactors in biological systems as the binding site for toxic gaseous molecules, such as NO and CO. The binding of these non-O_2_ gaseous molecules to the heme iron complex regulates various important physiological and pathological functions associated with O_2_ binding [9,10,11]. Artificial meat is one of the hottest topics in the field of food science; however, its acceptance is affected by its color. So, we hope to improve its color by using heme [12,13,14,15,16]. Heme can be extracted from animal blood by organic solvent or enzymatic hydrolysis; however, this is against animal welfare [16]. Biosynthesized heme can satisfy the need of color improvement as well as iron supplementation. Therefore, the animal-free and biotechnological production of free-heme molecules is essential for high-yield, economical and eco-friendly production.

The biosynthesis of heme has attracted extensive attention with the development of metabolic engineering and synthetic biology. Ge et al. [17] improved ALA and heme production by upregulating *hemB, hemG, hemH, hemAs* and *hemL* expression, and increasing the copy number of the genes *hox1* and *pcyA*. [18]. Feng et al. [19] improved heme production by regulating the expression of heme peroxidase in *E. coli*. The regulation of the heme production pathway has always been a concern due to the complex regulatory system of heme production.

5-aminolevulinic acid (ALA) is a universal metabolite in the biosynthesis of heme, that is synthesized by one of two different routes: the C4 pathway or the C5 pathway (Figure 1). The C4 pathway, also called the ‘Shemin pathway’, is present in mammals, fungi and α-proteobacteria. The C5 pathway of heme biosynthesis is found in plants, most bacteria and archaea. Compared to the C4 pathway, the C5 pathway uses simple carbon sources with higher titer of heme as substrates; therefore, it is more suitable for the biosynthesis of heme in prokaryotes [20,21,22,23]. In the C5 pathway, glutamate was firstly catalyzed by glutamyl-tRNA synthetase (*gltX*) and generate glutamyl-tRNA. Secondly, glutamyl-tRNA was catalyzed by glutamyl-tRNA reductase (*hemA*) to generate glutamate-1-semialdehyde, which is finally catalyzed by glutamate-1-semialdehyde transaminase (*hemL*) to generate ALA. This is followed by the production of bilirubinogen-by-bilirubinogen synthase (*hemB*). Subsequently, hydroxymethyl bile is produced by bilirubinogen deaminase (*hemC*). Then, the uroporphyrinogen III is produced by hydroxymethyl bile through the uroporphyrinogen III synthase (*hemD*). Coproporphyrinogen III is then generated by decarboxylase (*hemE*). Protoporphyrinogen IX is produced by coproporphyrinogen III oxidase (*hemF*). Protoporphyrin IX is then catalyzed by protoporphyrinogen oxidase (*hemG*) to produce protoporphyrin IX, and finally with Fe^2+^ to produce heme by the action of ferrous chelatase (*hemH*) [24].

In recent years, various hosts have been used to construct heme production strains, including *E. coli*, *Corynebacterium glutamicum* and *Salmonella typhimurium* [25,26,27,28,29]. Heme production can be effectively improved by enhancing the ALA production pathway, including accumulating succinyl CoA precursors [30], overexpressing the ALA transporter protein *RhtA,* a two-stage fermentation strategy [31], overexpressing transcriptional regulators *DtxR* [13]. Most bacteria (including *E. coli*) synthesize ALA and heme using glucose as a substrate by the C5 pathway, which not only simplifies the ALA and heme production pathway, but also saves costs and achieves higher yields [32,33,34]. *E. coli* is a universal model microorganism; in these studies, *E. coli* BL21(DE3) was mostly used as a host, and there is little study on heme-production potential of other kinds of *E. coli*.

In this study, we developed an engineered *E. coli* capable of de novo production of heme (Figure 1). To this end, we first screened for a host and a plasmid suitable for ALA and heme production. To improve the heme production, the genes *hemA* and *hemL* in the C5 pathway of ALA production were enhanced. Then, the genes *cysG* and *hemX*, that are involved in the precursor extracellular transport in heme production, were knocked out to improve the yield of heme from ALA. The gene *cyoE* was also knocked out to prevent heme conversion. Next, the Ccm and Dpp pathways were enhanced for extracellular transport of heme. Finally, fermentation optimization of recombinant *E. coli* was performed in 3-L bioreactors.

## 2. Results and Discussion

### 2.1. Improving HemA and HemL Expression in E. coli Host Strains

To screen a proper host for heme production in *E. coli* strains, the expression of genes *hemA* and *hemL* coding for glutamyl-tRNA reductase (GluTR) and glutamyl-1-hemal transaminase (GSA-AT) in the C5 pathway of ALA production were first enhanced in *E. coli* BL21(DE3). Plasmids pAC-*hemA*, pAC -*hemL* and pAC-*hemA*-*hemL* were constructed based on the plasmid pACYCDuet-1 (Appendix A). Recombinant strains EBL-pAC-A, EBL-pAC-L and EBL-pAC-AL were obtained by introducing plasmids pAC-*hemA*, pAC -*hemL* and pAC-*hemA*-*hemL* were constructed based on the plasmid pACYCDuet-1 into *E. coli* BL21(DE3), EJM-pAC-A, EJM-pAC-L, EJM-pAC-AL and strain EJM-pAC-K and strain EBL-pAC-K without enhancing target genes *hemA* and *hemL* target gene were constructed by using the techniques described in Section 3.3. After shake-flask fermentation, the recombinant strain EBL-pAC-AL provided 55.48 ± 4.69 mg/L heme in 8-h fermentation. The titer of EBL-pAC-AL was 3.71-fold that of recombinant EBL-pAC-A 14.95 ± 0.05 mg/L, and was 6.80-fold that of recombinant EBL-pAC-L 8.15 ± 1.07 mg/L. The titer of EBL-pAC-AL was 49.04 mg/L, 6.24-fold that of the strain EBL-PAC-K (Figure 2A). The above results indicate that the overexpression of *hemA* and *hemL* could significantly improve the ALA production. 

Then, the recombinant plasmid pAC-*hemA*-*hemL* was converted into *E. coli* NovaBlue(DE3), *E. coli* Turner(DE3), *E. coli* BL21(DE3) and *E. coli* JM109(DE3), resulting in recombinant strains ENO-pAC-AL, ETU-pAC-AL, EBL-pAC-AL and EJM-pAC-AL. The titer of EJM-pAC-AL was 3.03-fold that of recombinant ENO-pAC-AL 29.16 ± 4.70 mg/L, and was 2.75-fold that of recombinant ETU-pAC-AL 32.97 ± 1.66 mg/L. The titer of EBL-pAC-AL was 1.87-fold that of recombinant ENO-pAC-AL 29.16 ± 4.70 mg/L, and was 1.72-fold that of recombinant ETU-pAC-AL 32.97 ± 1.66 mg/L. These results indicated that the recombinant strains EJM-pAC-AL and EBL-pAC-AL had obvious advantages in ALA synthesis (Figure 2 and Figure 3A,B).

In order to screen for the highest quality plasmids here we selected three recombinant plasmids namely pAC-*hemA*-*hemL*, pCD-*hemA*-*hemL* and pET-*hemA*-*hemL*, which were simultaneously transformed into host *E. coli* BL21(DE3) and *E. coli* JM109(DE3) to obtain recombinant strains EBL-pAC-AL, EBL-pCD-AL, EBL-pET-AL, EBL-pET-AL, EJM-pAC-AL, EJM-pCD-AL and EJM-pET-AL. The highest ALA titer of recombinant strain EJM-pCD-AL was 90.55 ± 1.72 mg/L (Figure 3C,D). At the same time, the maximum production of heme was 4.34 ± 0.02 mg/L produced by EJM-pCD-AL. From this, the best recombinant plasmid pCD-*hemA*-*hemL* was screened for further study. Therefore, strain EJM-pCD-AL was selected for further study.

### 2.2. Increasing Heme Production by Removing Competitive Pathways

To remove the competitive pathways, we knocked out *CysG*, *hemX* and *CyoE* genes, respectively, in *E. coli* JM109(DE3), resulting in EJM-ΔCysG, EJM-ΔhemX and EJM-ΔCyoE (Appendix A). Then, the plasmid pCD-*hemA*-*hemL* was transformed into the three strains, resulting recombination strains EJM-ΔCysG-pCD-AL, EJM-ΔhemX-pCD-AL and EJM-ΔCyoE-pCD-AL, respectively. The fermentation was first carried out in shaking flasks and samples were taken at 4 h intervals to determine ALA and heme titer (Figure 4A–E). By observing the color of the fermentation broth, we found that the broth changed from light yellow to dark brown after fermentation for 48 h (Figure 4F). We obtained a heme titer of 9.06 ± 0.01 mg/L for strain EJM-ΔCysG-pCD-AL, and the extracellular heme secretion titer was 4.48 ± 0.01 mg/L, while the ALA titer decreased to 39.53 ± 1.38 mg/L, suggested that the knockout of the gene *CysG* could increase the heme titer [35] (Figure 4A). The heme production of strain EJM-ΔhemX-pCD-AL after knockout of the gene *hemX* was 8.62 ± 0.08 mg/L, the increase in heme titer was not obvious, at which point the ALA production did not change much (Figure 4B). We obtained from this result that knocking out the gene *hemX* had less effect on the increase in heme production, and also found that there was a certain loss of precursor material production in a certain range of heme production increase. We found *CyoE* knockout can greatly increase heme production while keeping ALA production almost unchanged (Figure 4C). At this point the recombinant strain EJM-ΔCyoE-pCD-AL reached a maximum titer of 9.43 ± 0.03 mg/L in shake flask culture, and had a 2.17-fold increase in heme production compared with the EJM-pCD-AL strain (Figure 4F). At this time, heme extracellular production reached 4.38 ± 0.04 mg/L. This suggested that blocking the pathway from heme-to-heme O could greatly accumulate heme production. Synthesis of Siroheme catalyzed by sirohydrochlorin ferrochelatase and uroporphyrin-Ⅲ C-methyltransferase (encoded by *CysG* and *hemX* genes, respectively) can be blocked. In addition, the gene *CyoE* which encodes heme O synthase, was knocked out to prevent the consumption of heme.

### 2.3. Overexpression of Heme Exporters to Increases Heme Production

In *E. coli*, the effect of Ccm pathway on heme production has been demonstrated [32], but the effect of Dpp pathway on heme production has never been studied. In this study, the regulation of *CcmA* and *DppA* genes influenced the output of heme protein to further observe the influence on ALA and heme production. The recombinant strain EJM-ΔCyoE-pCD-AL screened in the early stage was transformed into the competent cells obtained by gene editing technology to obtain strains EJM-Ccm-ΔCyoE-pCD-AL and EJM-Dpp-ΔCyoE-pCD-AL. By observing the color of fermentation broth, we found that the fermentation broth changed from light yellow to dark brown after 48 h of fermentation (Figure 4F). We guessed that the heme content might be increased to some extent at this time. After the analysis in Section 3.6, the heme titer of strain EJM-Ccm-ΔCyoE-pCD-AL was 8.94 ± 0.14 mg/L, and the extracellular heme secretion was 4.43 ± 0.08 mg/L. The heme titer of strain EJM-Dpp-ΔCyoE-pCD-AL was 9.04 ± 0.06 mg/L, and the extracellular heme secretion was 4.45 ± 0.08 mg/L. The EJM-Ccm-ΔCyoE-pCD-AL strain overexpressing the *CcmA* gene had a 2.05-fold increase in heme production compared with the EJM-pCD-AL strain. The EJM-Dpp-ΔCyoE-pCD-AL strain overexpressing the *DppA* gene had a 2.08-fold increase in heme production compared with the EJM-pCD-AL strain. This was consistent with our guess that there was indeed an increase in heme production compared to the previous strain EJM-pCD-AL (Figure 4F). At this time, the output of ALA decreased to a certain extent, the output of EJM-Ccm-ΔCyoE-pCD-AL was 58.01 ± 1.38 mg/L, and the output of EJM-Dpp-ΔCyoE-pCD-AL was 40.09 ± 1.71 mg/L (Figure 4B). ALA and heme production of some recombinant strains can be seen in Table 1 and Appendix A.

### 2.4. Improving the Heme Production in Bioreactor

#### 2.4.1. Optimization of Fermentation Conditions

On the basis of the TB medium, we optimized the concentration of carbon source glucose (Glc), the trace element Fe^2+^ and the precursor glutamate (Glu), which were all required in the C5 pathway. The specific added amount can be seen in Section 3.5. As shown in (Figure 5), the increasing concentration of glucose, FeCl_2_ and glutamate could promote the accumulation of heme and ALA. The effect of glucose was the most obvious, which further increased the titer of heme to 10.95 ± 0.07 mg/L (Figure 5). Fe^2+^ was used as a metal ion chelated by heme, and the content of exogenous Fe^2+^ increasing within a certain range was also beneficial to the synthesis of heme. Glutamate, as the precursor of *E. coli* C5 pathway synthesis, can promote the accumulation of heme, and the promotion of ALA synthesis was very obvious. After optimization of the fermentation medium, the final heme titer reached up to 10.95 ± 0.07 mg/L and the ALA titer reached up to 111.98 ± 1.9 mg/L (Figure 5).

#### 2.4.2. Fed-Batch Fermentation of Heme by Recombinant Strains

To test the high-level ALA and heme production potential of engineered strains, fed-batch fermentations were conducted in 3-L bioreactors using the optimized producer strain EJM-∆CyoE-pCD-AL and the primary producer strain EJM-pCD-AL as a control. Overall, strain EJM-∆CyoE-pCD-AL was significantly better than strain EJM-pCD-AL in terms of cell growth, heme production and ALA production (Figure 6). After 48 h for fermentation, 303 ± 1.21 mg/L of ALA was produced by strain EJM-∆CyoE-pCD-AL, representing a 39.6% improvement compared to strain EJM-pCD-AL (224.17 ± 3.08 mg/L) (Figure 6). At the same time, 28.20 ± 0.77 mg/L of heme was produced by strain EJM-∆CyoE-pCD-AL, representing a 35.2% improvement compared to strain EJM-pCD-AL (20.27 ± 0.17 mg/L) (Figure 6). The above results indicate that by enhancing the expression of key genes *hemA* and *hemL*, and simultaneously knocking out the gene *CyoE* that decomposes ALA, played an important role in the efficient production of heme. This also provided an effective strategy for the industrial production of heme.

## 3. Materials and Methods

### 3.1. Strains and Plasmids

All plasmids and bacterial strains constructed and used in this study are listed in Appendix A. *E. coli* DH5α was used as template for the amplification of target genes *hemA* and *hemL*. *E. coli* JM109 was used for plasmid construction and preservation. *E. coli* JM109(DE3), *E. coli* BL21(DE3), *E. coli* Turner(DE3) and *E. coli* NovaBlue(DE3) were used as host for ALA and heme production. The genomic DNA of *E. coli* JM109 was used for the amplification of *CysG*, *CyoE*, *HemX*, *CcmA* and *DppA*. Plasmids pKD13 and pKD46 were used for genome editing of *E. coli*.

### 3.2. Culture Conditions

For plasmids construction, strains were cultured in a 250 mL shake flask containing 50 mL of Luria-Bertani (LB) medium (10 g/L tryptone, 5 g/L yeast extract, 10 g/L NaCl) at 37 °C 220 rpm. The solid medium was prepared by adding 15 g/L agar to the liquid LB medium. Chloramphenicol (20 μg/mL), streptomycin (100 μg/mL), ampicillin (100 μg/mL) and kanamycin (100 μg/mL) were added to the medium depending on the situation. Single colonies harboring different plasmids were picked up and cultured in 5 mL LB media at 37 ℃ overnight with continuous shaking. Then, 5.0% inoculum of the seed was transferred to a 250 mL flask with about 50 mL Terrific Broth (TB) medium (11.8 g/L tryptone, 23.6 g/L yeast extract, 9.4 g/L K_2_HPO_4_, 2.2 g/L KH_2_PO_4_, 4 mL glycerol) for ALA and heme production. All recombinant strains were cultured at 37 °C, 220 rpm. Chloramphenicol, streptomycin, ampicillin, or kanamycin was used for plasmids selection. Gene expression was induced with initial addition of isopropyl-β-D-thiogalactopyranoside (IPTG, 0.1 mM).

For genome modification, SOB medium (20 g/L tryptone, 5 g/L yeast extract, 0.5 g/L NaCl, 5 g/L MgSO_4_·7H_2_O) as a nutritionally rich growth medium was used for the preparation of competent cells to improve the transfection efficiency. In addition, L-arabinose (10 mM) was also added to induce the expression of Red-recombinase. SOC medium (20 g/L tryptone, 5 g/L yeast extract, 0.5 g/L NaCl, 5 g/L MgSO_4_·7H_2_O, 3.6 g/L D-Glucose) was used in the last stage of transformation. As plasmid pKD46 is temperature sensitive, cells used for genome editing should be cultured at 30 °C.

### 3.3. Construction of Recombinant Plasmids and Strains

Standard molecular genetic techniques were used for DNA manipulation [36]. The recombinant plasmids and strains constructed and used in this study are listed in Appendix A. The primers used for constructing plasmid and Red-based recombineering are listed in Appendix A. The genomic DNA of *E. coli* DH5α was used to amplify the *hemA* and *hemL*. pACYCuet-1, pCDFDuet-1 and pETDuet-1 vectors were used for *E. coli* for the overexpression of target genes. H*emA* and *hemL* genes were amplified using primers hemA-F/hemA-R and hemL-F/hemL-R to construct plasmid pAC-*hemA*-*hemL*, pCD-*hemA*-*hemL* and pET-*hemA*-*hemL*. The linear plasmid and coding sequence were assembled using the ClonExpress II One Step Cloning Kit (Vazyme, Nanjing, China). The recombinant plasmids pAC-*hemA*-*hemL*, pCD-*hemA*-*hemL* and pET-*hemA*-*hemL* were then transformed into host receptor cells *E. coli* JM109(DE3), *E. coli* BL21(DE3), *E. coli* Turner(DE3) and *E. coli* NovaBlue(DE3), respectively, to construct recombinant strains EBL-pAC-AL, EBL-pCD-AL, EBL-pET-AL, EJM-pAC-AL, EJM-pCD-AL, EJM-pET-AL, ENO-pAC-AL, ENO-pCD-AL, ENO-pET-AL, ETU-pAC-AL, ETU-pCD-AL and ETU- pET-AL.

### 3.4. Red/ET-Based Recombineering

In order to knock out the genes *CysG*, *CyoE* and *HemX* on the *E. coli* genome, the Red/ET homologous recombination system was applied here to design three knockout frames consisting of two homologous regions (one 550 bp upstream and one 500 bp downstream of the gene to be knocked out) and one region replacing the pKD13 resistance gene fragment of the gene to be knocked out [37]. These three regions were amplified and purified using the primers in Appendix A (kanR-F/R, kanR-YZ, cysG-up-F/R, cysG-down-F/R, cyoE-up-F/R, cyoE-down-F/R, hemX-up-F/R and hemX-down-F/R) and then subjected to fusion polymerase chain reaction (PCR). To enhance the expression of the heme extracellular transport pathway, two insertion frames were designed here by the Red/ET homologous recombination system, where the primers from Appendix A (kanR-F/R, kanR-YZ, dppA-up-F/R, dppA-T7-F/R, dppA-F/R, ccmA-up-F/R, ccmA-T7-F/R and ccmA-F/R) were separately amplified and purified for fusion the insertion frames CcmAup-KanR-T7-CcmA and DppAup-KanR-T7-DppA were obtained by polymerase chain reaction (PCR).

### 3.5. Fed-Batch Fermentation

The culture used in fed-batch fermentation was optimized based on TB medium with different concentrations of glucose (30 g/L, 40 g/L and 50g/L), FeCl_2_ (5 mg/L, 7 mg/L and 9 mg/L) and glutamate (0.4g /L, 0.6 g/L and 0.8 g/L).

Fed-batch fermentations for ALA and heme production were performed as follows: 40 mL seed of the engineered strain cultured in LB medium at 37 °C, 220 rpm for 16 h was inoculated to 3-L fermenter with approximately 1.5 L of fermentation medium (11.8 g/L tryptone, 23.6 g/L yeast extract, 9.4 g/L K_2_HPO_4_, 2.2 g/L KH_2_PO_4_, 6 mL glycerol, 40 g/L glucose and 7 mg/L FeCl_2_). The culture conditions were maintained as follows: temperature 37 °C, airflow rate 1.0 vvm, agitation speed 500 rpm. pH was automatically controlled at 7.0.

### 3.6. Analytical Methods

Cell growth was measured at OD_600_ using a UV-vis spectrophotometer (Yuanxi Co, Shanghai, China) after the culture was diluted to a proper volume with distilled water. ALA concentration was measured using the colorimetric called Ehrlich’s reagent. Take 1 mL cultured cells were centrifuged at 12000 rpm for 5 min. An amount of 300 μL of the supernatant was chemically reacted with 400 μL of sodium acetate buffer (PH 4.6) and 35 μL acetylacetone at 100 °C for 15 min. After cooling to room temperature, 440 μL Modified Ehrlich’s reagent was added for 10 min and the absorbance value was measured at 554 nm using a spectrophotometer [38]. For the measurement of intracellular heme, 1 mL cultured cells were centrifuged at 4000 rpm for 6 min. After separating supernatant, the cell pellet was disrupted using-modified actone: HCl extraction methods described by Espinas et al. [39]. After 1mL of acetone: HCl (95:5) buffer was added to the cell-harvested tube, the mixture was vortexed and diluted with 1 mL of 1 M NaOH. The intracellular sample was disrupted, and the supernatant was filtered using an MCE filter for concentration analysis. For the measurement of extracellular heme, the supernatant from the cell culture was mixed with 1 M NaOH at a 1:1 ratio and filtered using an MCE filter for concentration analysis. Heme concentration was determined using a high-performance liquid chromatography (HPLC) system (Agilent Co, Palo Alto, CA, USA). The filtered sample was separated in a Symmetry^®^ C18 HPLC Column 5 μm particle size, 4.6 × 250 mm. Solvent A is a 10:90 (*v/v*) HPLC grade methanol: acetonitrile mixture, and solvent B is a 0.5% (*v/v*) trifluoroacetic acid (TFA) in HPLC grade water. The flow rate was 0.6 mL/min for 40 min, and the absorbance was determined at 405 nm [13,20]. The HPLC chromatograms of heme were shown in Appendix A. In this experiment, differences of two groups of data were determined by a two-tailed Student’s *t* test, and the statistical significance is indicated as * for *p* < 0.05 and ** for *p* < 0.01.

## 4. Conclusions

In this study, we engineered *E. coli* for improving the production of ALA and heme. Here we chose the C5 pathway of heme biosynthesis and demonstrated that co-expression of the genes *hemA* and *hemL* can significantly promoted the accumulation of ALA and heme. We also knocked out the genes *CysG*, *hemX* and *CyoE*, which may prevent intracellular degradation of ALA hinder heme biosynthesis, and found that knocking out *CysG* was effective in further increasing heme accumulation, and *CyoE* knockout can greatly increase heme production while keeping ALA production almost unchanged. Finally, membrane engineering was attempted by enhanced putative heme exporters. Ths enhanced the dipeptide transport system substrate-binding protein and heme exporter protein A of heme outward transport regulated by the genes *DppA* and *CcmA*. It was demonstrated that overexpression of the genes *CcmA* and *DppA* encoding heme export proteins also enhanced heme production. The results of this study improved the heme production and its yield from ALA by preventing the degradation of ALA and heme and enhancing the heme transport. However, the titer and yield of heme was still not qualified for large-scale production, therefore, further modifications are needed to maximize the potential of the engineered strains to fit the industrial applications.

## Figures and Tables

**Figure 1 ijms-23-15524-f001:**
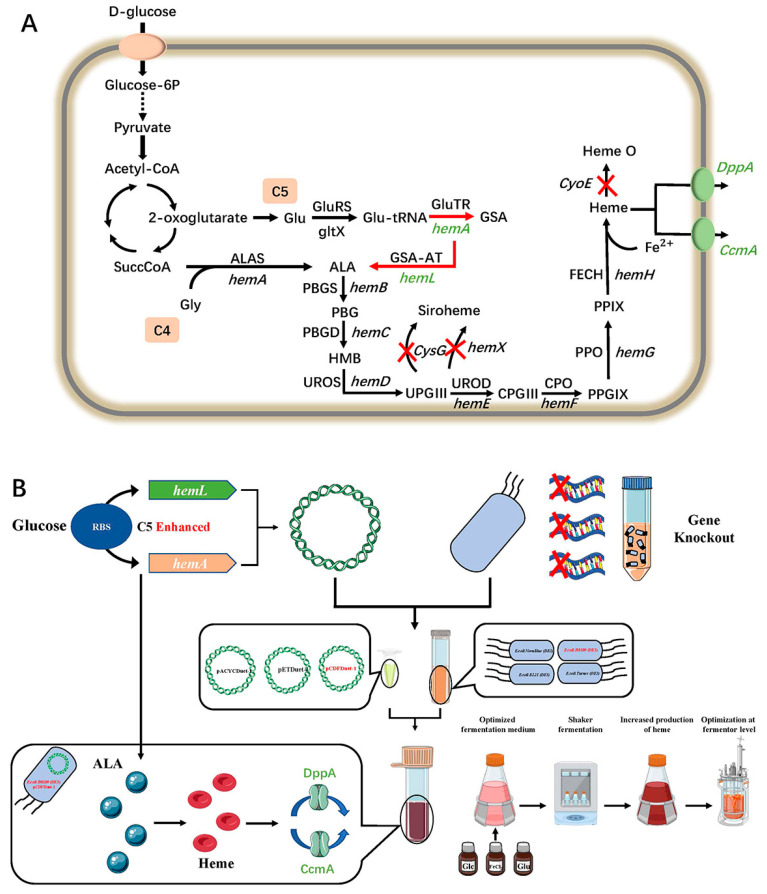
(**A**) A schematic overview describing the target pathway and overall metabolic strategies for ALA and Heme production in *E. coli*. In this study, glucose was the only major source of energy for cell growth and porphyrin production. The green letter represents the gene overexpression in *E. coli*. The red X-shape denote the gene knockout. The solid-line arrow and dashed-line arrow indicate the general metabolic pathway and abbreviated pathway, respectively. The red arrow denotes enhanced metabolic flux by the combinatorial overexpression of *hemA* and *hemL*. Green ovals indicate enhanced extracellular transport pathways for heme by insertion of the target gene. (**B**) Strategies for modifying the biosynthesis of heme in *E. coli*.

**Figure 2 ijms-23-15524-f002:**
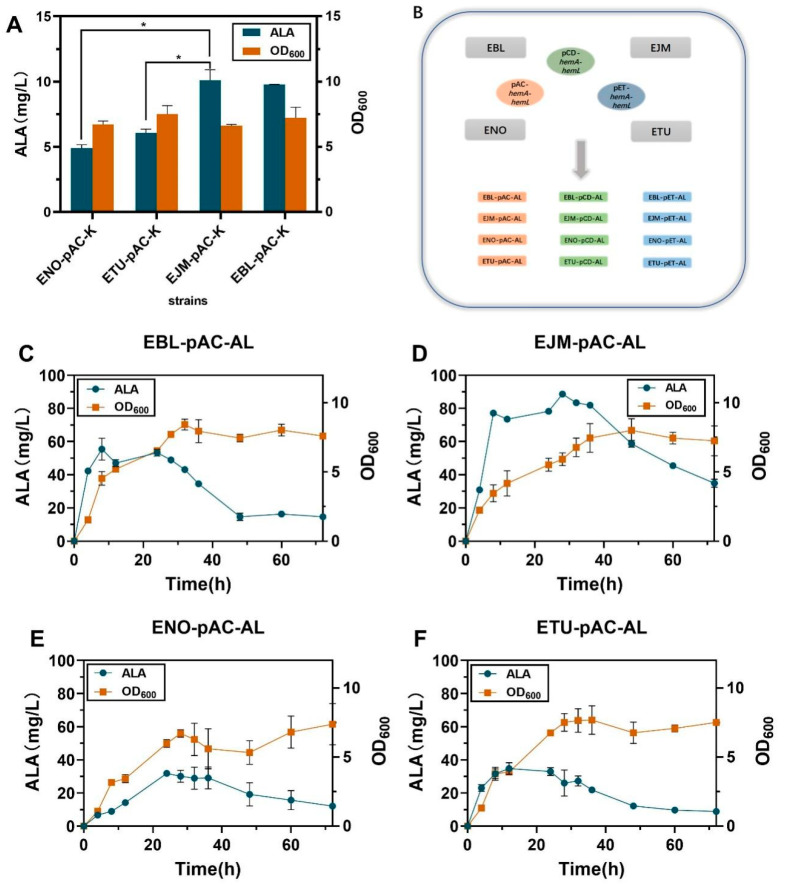
(**A**) OD_600_ and ALA production of the no-loaded strains at 36 h of fermentation. (**B**) Recombinant plasmid construction flowchart. (**C**–**F**) Growth curve and ALA production of recombinant strain fermented for 72 h. (The recombinant strains corresponding to Figure 2C–F are EBL-pAC-AL, EJM-pAC-AL, ENO-pAC-AL, ETU-pAC-AL, respectively). All data indicate the mean of two independent biological experiments and error bars represent standard deviation. The symbol * stands for and *p* < 0.05, as determined by Student’s *t*-test.

**Figure 3 ijms-23-15524-f003:**
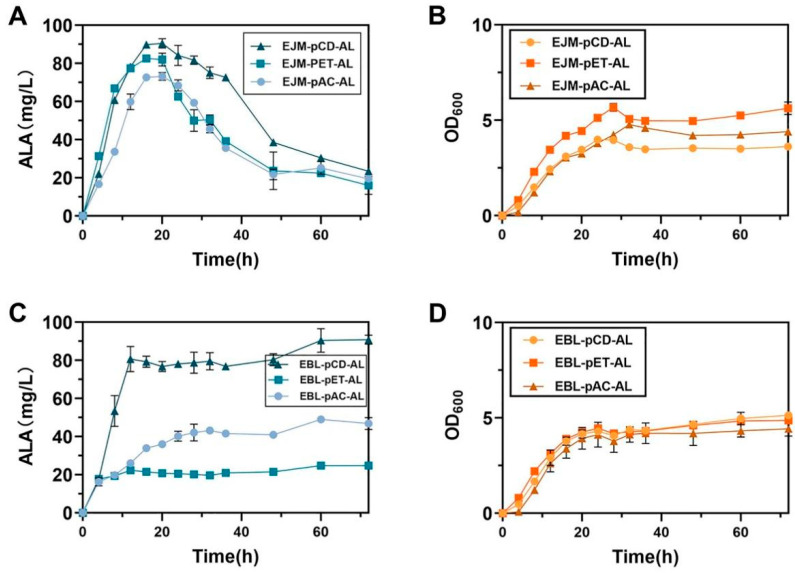
(**A**) The recombinant plasmids pCD-*hemA*-*hemL*, pET-*hemA*-*hemL* and pAC-*hemA*-*hemL* were transformed into the host *E. coli* JM109(DE3) for ALA production after 72 h of fermentation. (**B**) Growth curves of recombinant plasmids pCD-*hemA*-*hemL*, pET-*hemA*-*hemL* and pAC-*hemA*-*hemL* were transformed into the host *E. coli* JM109(DE3) for 72 h fermentation. (**C**) The recombinant plasmids pCD-*hemA*-*hemL*, pET-*hemA*-*hemL* and pAC-*hemA*-*hemL* were transformed into the host *E. coli* BL21(DE3) for ALA production after 72 h of fermentation. (**D**) Growth curves of recombinant plasmids pCD-*hemA*-*hemL*, pET-*hemA*-*hemL* and pAC-*hemA*-*hemL* were transformed into the host *E. coli* BL21(DE3) for 72 h fermentation. All data indicate the mean of two independent biological experiments and error bars represent standard deviation.

**Figure 4 ijms-23-15524-f004:**
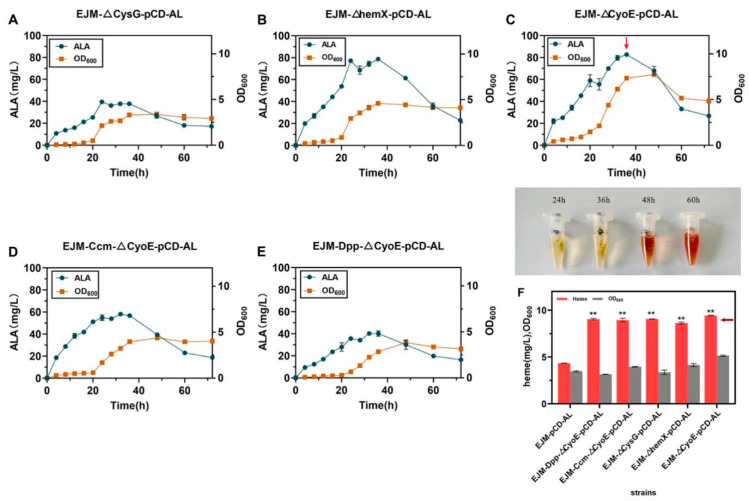
(**A**–**E**) The growth curve and ALA production of the gene-edited strain within 72 h of fermentation. (**F**) The color change in fermentation broth at different stages of fermentation and heme titer of different recombinant strains. All data indicate the mean of two independent biological experiments and error bars represent standard deviation. The symbol ** stands for and *p* < 0.01, as determined by Student’s *t*-test.

**Figure 5 ijms-23-15524-f005:**
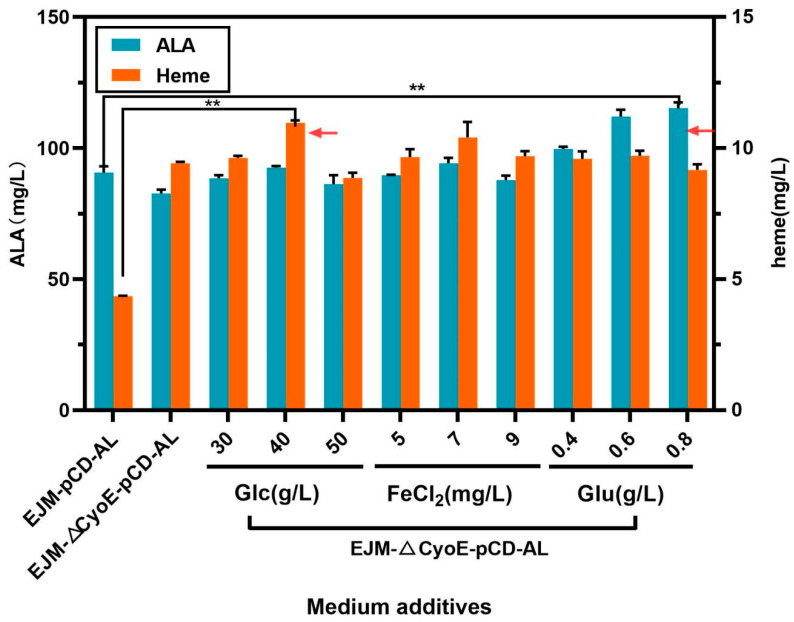
Heme and ALA production of recombinant strain EJM-∆CyoE-pCD-AL under medium-optimized conditions. The abscissa shows the heme and ALA production of control strain EJM-pCD-AL and the yield of recombinant strain EJM-∆CyoE-pCD-AL under optimized conditions, respectively. The symbol ** stands for and *p* < 0.01, as determined by Student’s *t*-test.

**Figure 6 ijms-23-15524-f006:**
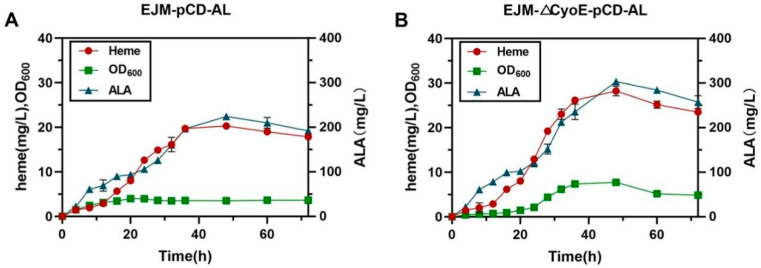
Fed-batch fermentation results of engineered strains. (**A**) Fed-batch fermentation profiles of the strain EJM-pCD-AL during 72 h. (**B**) Fed-batch fermentation profiles of the strain EJM-△CyoE-pCD-AL during 72 h. The red circle represents heme production, the green square represents OD_600_ and the blue triangle represents ALA production.

**Table 1 ijms-23-15524-t001:** This table shows the ALA and heme production detected by fermentation of recombinant strains constructed in this study.

Strain	Characterization	ALA (mg/L)	Heme (mg/L)
EBL-pCD-AL	E. coli BL21(DE3) contains pCD-hemA-hemL	90.47 ± 4.37	3.82 ± 0.12
EJM-pCD-AL	E. coli JM109(DE3) contains pCD-hemA-hemL	90.55 ± 1.72	4.34 ± 0.02
EJM-Dpp-△CyoE-pCD-AL	Recombinant strains overexpressing the Dpp pathway by gene editing after knockout of gene CyoE	40.09 ± 1.71	9.04 ± 0.06
EJM-Ccm-△CyoE-pCD-AL	Recombinant strains overexpressing the Ccm pathway by gene editing after knockout of gene CyoE	58.01 ± 1.38	8.94 ± 0.14
EJM-△CysG-pCD-AL	Recombinant strain after knockout of gene CysG	39.53 ± 1.38	9.06 ± 0.01
EJM-△hemX-pCD-AL	Recombinant strain after knockout of gene hemX	78.68 ± 0.83	8.62 ± 0.08
EJM-△CyoE-pCD-AL	Recombinant strain after knockout of gene CyoE	82.73 ± 0.98	9.43 ± 0.03

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
