# Peer review of "Efficient De Novo Biosynthesis of Heme by Membrane Engineering in Escherichia coli"

_ijms, 2022, doi:10.3390/ijms232415524_

Round 1

Reviewer 1 Report

The present work describes successful overexpression of heme by manipulating genes arrangements involved in the heme biosynthesis. The experiments are well designed and the results are solid and reliable. The manuscript is well written and readable. I think that this interesting work would certainly advance our understanding of the overexpression of heme in E. coli.  I recommend this paper for publication in the Journal. However, I raise some concerns that need to be addressed before publication. If those concerns are adequately address in the revised manuscript, this interesting report would be significantly strengthened.

Concerns that need to be addressed before publication.

[1] I think references are biased rather to food sciences. I would suggest the authors to add brief comments covering research fields of heme and heme-bound proteins in the first paragraph of Introduction by citing the following papers for general readers to understand the important role of heme. This is essential for publication of the present interesting paper.

Chem. Rev2015, 115, 6491.  Heme-based gas (O2, NO, CO) sensors.

Crit. Rev. Oncol. Hemato2018, 126, 121. Heme toxicity.

Chem. Soc. Rev. 2019, 48, 5624. Heme-responsive sensors.

Free Rad. Biol. Med. 2019, 133, 88. Heme movement.

JACSAu 2021, 9, 1296. Various heme proteins.

BBA Mol.Cell Res2021, 1868, 118881. Heme trafficking. 

Coord. Chem. Rev2022, 452, 214286. Bacterial heme biosynthesis.

Coord. Chem. Rev. 2022, 472, 214793. Heme acquisition.

Please let me remind that the present journal’s name is “…Molecular Sciences”.  Short comments about heme from “the Molecular/Chemical Science’s points of view” citing those papers would be beneficial/useful to significantly improve this novel paper.

[2] Overexpression in E. coli of many heme-proteins generates already-heme-bound proteins, but does not need to overexpress the heme iron complex itself separately.  How do the authors rationalize the present study? Also please explain the superiority of the present method compared to extraction of heme from bloods and meats. 

[3] Heme is not soluble in water. How did the authors address this problem? Is heme incubated/overexpressed into the cell membrane? Title says “… by Membrane Engineering…”, but I don’t see how membrane engineering works in the manuscript. Please emphasize this somewhere or in Conclusion. 

[4] I would suggest the authors to make a new figure describing genome organization of genes involved in heme biosynthesis.  Please describe cartoon depicting general features composing heme-related genes such as hemA, hemB, hemC, hemD, hemE, hemeF, hemG, hemH, hemL hemX, CyoE, CysG, CcmA, DppA (discussed in the text) in order to rationally and clearly explain relationships among those genes for general readers to grasp the points. Also please visually explain your experiments and results in this figure in order to clearly demonstrate the present important findings. Figs. 1 and 2B are not enough. 

[5] Papers of heme and heme-proteins always provide absorption spectra. I don’t see any optical absorption spectra (340-700 nm) of cells overexpressing Fe3+ heme complex under various conditions and those of purified (semi-?) Fe3+ heme complex in the present paper. Please incorporate spectra into the Supplementary Information. 

[6] I would suggest the authors to consider to use Supplementary Information in order to simplify the main text and to emphasize important points in the main text about the present interesting study. For example, Table 1 might be moved to Supplementary Information. Also, some of experimental results shown in Table 2 and Fig. 2 might be moved to Supplementary Information for the same reasons.  

In summary, the present paper is interesting/novel and should be published in the Journal. If some concerned are adequately addressed in the revised version, the present superb paper would be further improved. 

Reviewer 2 Report

This paper presents extensive genetic engineering of E. coli strains with the aim of enhancing biosynthesis of heme. The authors do a good job of describing a rather comprehensive and complex study with clarity and brevity. The results presented offer a significant improvement on previous efforts and seem to be a strong step toward efficient industrial-scale biosynthesis of heme. Minor edits of the English are needed, as there are several incomplete sentences, verb tense errors, or other minor grammatical errors. The only technical criticism I have to offer is that the presentation of error bars representing standard deviation is inappropriate for this case. The data presented are largely averages of two replicates. When small data sets are averaged, calculation of a standard deviation is meaningless. Instead, it is appropriate to report the uncertainty of the mean. The error bars in all figures should be changed to reflect this value rather than standard deviation. 
